# Effect of Music Therapy as a Non-Pharmacological Measure Applied to Alzheimer's Disease Patients: A Systematic Review

**E. Begoña García-Navarro** [1,2,*] **, Ana Buzón-Pérez** [3] **and María Cabillas-Romero** [4]

1. Department of Nursing, University of Huelva, 21007 Huelva, Spain
2. ESEIS Research Group, COIDESO Research Center, University of Huelva, 21007 Huelva, Spain
3. Huelva-Costa Sanitary District, Andalusian Public Health System, 21005 Huelva, Spain
4. Department of Social Anthropology, Basic Psychology and Public Health, University Pablo de Olavide, 41013 Seville, Spain
* Correspondence: bego.garcia@denf.uhu.es or esperanza.garcia@denf.uhu.es

**Abstract:** The objective of this research was to learn about the effects of music as a non-pharmacological therapeutic intervention applied to Alzheimer's disease. To this aim, we evaluated its results regarding symptomatology and caregiver burden. Methods: This systematic review followed the Preferred Reporting Items for Systematic Reviews and Meta-Analyses (PRISMA) criteria. We searched Pubmed/Medline (NLM), Web of Science, Scopus, Cochrane and Google Scholar, including articles from 1 January 2011–20 July 2021, using the keywords "Alzheimer's disease", "Music therapy", "Caregivers overload", "Amnesia retrograde" and "Clinical evolution". To select the articles our criteria included complete text availability, quantitative research of an experimental nature and studies which, at least, figured in SJR or in JCR. Results: We found a relationship between the application of music therapy in patients with Alzheimer's disease and an improvement regarding symptomatology, as it smoothed down the progress of the disease. Using music therapy in Alzheimer's patients also led to a decrease in caregivers' burden and an increase in their well-being. Conclusions: Our results showed the benefits of music therapy, as it improved both patients' symptomatology and caregivers' burden.

**Keywords:** Alzheimer's disease; music therapy; caregivers' overload; clinical evolution; retrograde amnesia

## 1. Introduction

According to the WHO [1], 47 million people in the world suffer from dementia, a chronic and progressive disorder. Dementia affects orientation, comprehension, memory, thinking, calculation, learning ability, language and judgment. Therefore, it is characterized by damages in terms of cognitive function. The symptoms of dementia are usually preceded by a deterioration in emotional control, social behavior and motivation, which are not related to physiological aging processes. Among the explaining factors connected to this disorder we can find brain injuries and several diseases [2].

This syndrome has a remarkable incidence rate, since every year about 10 million new patients are diagnosed worldwide. Additionally, we need to consider the enormous global cost it involves, which is more than EUR 867 billion per year and is expected to double in 2030. Another fact to consider is the enormous social burden that this disease implies regarding the biopsychosocial and labor spheres. Caregivers are affected in more than 50% of cases and, consequently, additional support from health systems is required [3,4]. An increased longevity among the population has also caused an increase in the prevalence of chronic diseases, which include chronic mental disorders such as dementia and one of its subtypes, namely Alzheimer's disease [5].

According to the International Society of Neuropathology "Alzheimer's disease is a neurodegenerative disorder characterized clinically by dementia and pathologically by the extracellular accumulation of peptide AB in the parenchyma and by the intraneuronal

accumulation of tau protein in neurofibrillary tangles" [6]. This translates into a very varied symptomatology that sometimes makes it difficult to diagnose Alzheimer's disease, although its most prominent symptom is the onset of cognitive decline. Despite this definition, it is interesting to understand this disease going beyond its biological dimension. Disorders related to recent memory functioning are a distinctive feature of mild cognitive impairment, therefore leading to the suspicion of Alzheimer's disease [6,7]. Taking this into consideration, it is important to be aware of these signs to avoid their identification with normal processes related to aging. Doing so would facilitate that medical support is sought in the early moments of the disease, which would contribute to a slowdown in its development. The benefits of identifying early signs emphasize the importance of developing more specific diagnostic criteria for Alzheimer's, which would help early diagnosis and the initiation of treatment.

Although it is possible to pharmaceutically stabilize Alzheimer's symptomatology and smooth down its course, there is no cure or possible way to reverse its evolution. Additionally, there are no definitive non-pharmacological measures that increase the quality of life of people with Alzheimer's disease, nor that of their caregivers and relatives. This has important socioeconomic consequences for health systems and for the psychological, physical and cultural spheres of both patients and caregivers all over the world [5–8].

Music therapy is one of the non-pharmacological therapies that have proven to be valid to reduce some symptoms in Alzheimer's disease, even improving the feeling of well-being of patients [8]. Etymologically, music therapy means "therapy through music". Its origins date back to the second half of the nineteenth century where we find examples such as Dr. Rodríguez-Méndez who began to use music as a therapeutic treatment, or Dr. Vidal-Careta who, in 1882, wrote the first doctoral thesis relating music and medicine under the title "Music in its relations with medicine" [5]. In this thesis, he concludes that music helps to rest and represents a distraction, being also a moralizing social component. He also states that it is appropriate to use music in neurosis and in states of excitement or nervousness [6–8]. Recent studies [9–11] have reaffirmed the benefits of music therapy on mental health and motivation in both patients with Alzheimer's disease and their caregivers. To explore how music therapy improves the symptomatology of dementia in the long term, Ledger and Baker's study [9] observed the evolution and time in cases and a control group. The main results of their study refer to music therapy as improving and benefitting patients with regard to behavioral disorders and emotional stability. Additionally, they confirm that music therapy enhances positive emotions in patients and reduces the caregiver's burden.

Music therapy acts as a distraction and helps to deal with emotional and affective problems, motivating patients to live longer [10]. In addition, this therapy facilitates sharing experiences with other people, helps maintaining contact with reality at all times, and contributes to the prevention of complications [11]. Music therapy complements pharmacological treatments, increasing the effectiveness of these treatments in general.

Music therapy can be active or passive. In its active format, patients sing songs and can create music. In the passive modality, patients listen to music. Currently, music therapy programs are being developed and applied in patients with neurological problems related to dementia, Parkinson's, Alzheimer's and strokes provoked by cerebrovascular accidents [11].

The purpose of this literature review was to analyze and provide scientific evidence with the objective of assessing whether the symptoms of Alzheimer's can be made more bearable with non-pharmacological therapies, specifically focusing on music therapy.

## 2. Materials and Methods

Our research question addressed the effects of music therapy as a non-pharmacological therapy in patients with Alzheimer's disease. The main objective of this review was to examine the current scientific evidence about the effects of music therapy as a non-pharmacological therapy applied in these patients. Specifically, we wanted to explore the influence of music therapy on Alzheimer's disease symptoms and its effects on caregivers' burden.

This systematic review followed the preferred Reporting Items for Systematic Reviews and Meta-Analysis (PRISMA) criteria [12]. On the other hand, in order to evaluate the methodological quality of the documents found and to confirm the selection of suitable works, the Critical Appraisal Skills Program (CASPe) was used to analyze systematic reviews (Cabello López, 2015) and randomized controlled trials (Cabello, 2005) [13]. The Queen's Joanna Briggs Collaboration scale (Joanna Briggs Institute Levels of Evidence and Grades of Recommendation Working Group, 2013) was also used for the analysis of descriptive studies [14,15].

A focused systematic literature search strategy was developed by the research team to identify relevant articles. The search strategy used a combination of medical subject headings (MeSH) and titles and abstract keywords. The search strategy was the most appropriate according to the objectives proposed in this bibliographic review, using six electronic databases: Pubmed/Medline (NLM), Web of Science, Scopus, Cochrane and Google Scholar, including articles from 1 January 2011–20 July 2021.

Additionally, synonyms for keywords and other search terms were used to ensure the search was as comprehensive as possible (Table 1).

**Table 1.** Terms used in the search.

| MeSH | Terms |
| --- | --- |
| Alzheimer disease | Alzheimer dementia, retrograde amnesia, clinical evolution |
| Music therapy | Sensory art therapies |
| Caregiver burden | Caregiver overload |

The search strategy was developed in PubMed (Table 1) and adapted for the other databases. To achieve these objectives, three search strategies were used combining the chosen descriptors with the Boolean operators AND and OR. Two of them were not effective in all the databases and were therefore excluded: "Alzheimer disease" AND ("Music therapy" OR "Caregiver overload" OR "clinical evolution"), and "Alzheimer disease" AND ("Music therapy" OR "Caregiver overload" OR "Retrograde Amnesia"). The definitive search strategy that was selected was "Alzheimer disease" AND ("Music therapy" OR "Caregiver overload" OR "Retrograde Amnesia" OR "clinical evolution").

*2.1. Inclusion and Exclusion Criteria*

Articles that met the following requirements were included:

_ Research published from 2013 to the present.
_ Experimental and interpretive studies of a qualitative and quantitative nature.
_ Languages: English, Spanish or Portuguese.
_ Articles that answered the Research Question.
_ Articles that covered the specific objectives.

Articles presenting the following circumstances were excluded:

_ They were repeated in different consulted sources.
_ Articles that did not appear in SJR (Scimago Journal and Country Rank) and/or JCR (Journal Citation Reports), and therefore did not ensure the required quality standards. The search strategy must have responded to the needs of the investigation, leading to a broad scenario in order to provide enough feedback to be able to argue the set objectives. Three search strategies were designed, each of them combining the chosen descriptors with the Boolean operators AND and OR.

*2.2. Search Strategy*

Our definitive search strategy was "Alzheimer disease" AND ("Music therapy" OR "Caregiver overload" OR "Amnesia retrograde" OR "clinical evolution"). It was the one

that best allowed us to specify, refine and approach the topics being discussed according to the objectives proposed in this bibliographic review using the selected databases.

After the bibliographic search, we turned to considering the quality of the articles with the CASPe program's Critical Evaluation Skills Program to analyze the systematic reviews (Cabello López, 2015) and randomized controlled trials (Cabello, 2005) and the Queen's Joanna Briggs Collaboration scale (Joanna Briggs Institute Levels of Evidence and Grades of Recommendation Working Party, 2013) for the analysis of the descriptive studies [13–15], as well as the criteria specified by SJR and JCR. Regarding JCR, the impact factor and the JCR quartile of the year in which the article was published were located through the Fecyt JCR tool. In the case of SJR, we used the SJR website to determine the impact factor and the quartile. Those articles that did not appear in either of these two tools were excluded. This led us to consider sixteen articles, which can be examined in Table A1 in the Appendix A. Once selected, we proceeded to an exhaustive reading, after which we extracted the names of the authors and classified the articles according to the objectives and their specific theme.

Methodologically, the sample size of the studies was extracted, and a classification was made, taking into account whether the study presented a control group or not. Several variables were used to classify the results and to extract the most relevant data from the articles. Two of our classification criteria were the place and the year in which the studies took place. Other data extracted from the articles for their classification regarded the therapy to which patients were subjected: music therapy, both passive and active, or other non-pharmacological therapies. An extraction of the main findings of each of the studies was performed.

Regarding the sample size of each study, a classification was made according to the stage and years of evolution of the disease, the place of residence, patients' age and the type of non-pharmacological therapies which they were subjected to, i.e., music therapy or others. In a percentage of the sample, the authors did not specify the stage or years of evolution of the disease. Another classification criterion was the non-specification of the named variables, since the authors did not particularize either the place of residence of the patients or their age. (Figure 1)

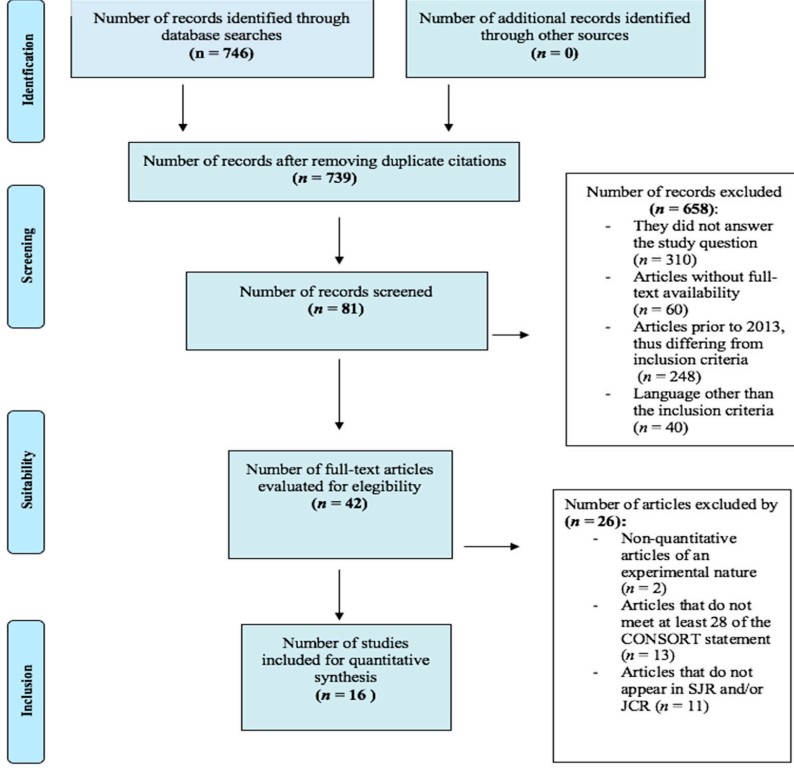

**Figure 1.** PRISMA flowchart (resource: own elaboration).

## 3. Results

### 3.1. Effect of Music Therapy on Caregivers' Burden

The effects of music therapy on caregivers' burden have received less attention than the effects on Alzheimer's symptomatology. In our study, only four out of sixteen articles took the issue into consideration, which can be explained by the fact that the effects of music therapy on the symptoms manifested by the patients aroused more interest among the researchers than the effects related to caregivers' burden. Considering that caregivers are an essential pillar of support in the process of Alzheimer's disease, it would be necessary to develop research in order to expand the existing literature in this field.

An experimental study carried out in France [16] designed with no control group compared two groups relating the efficacy of two non-pharmacological treatments in patients with Alzheimer's and their caregivers in a nursing home. In one group, patients received music therapy, whereas in the other group they organized culinary activities in which the caregivers participated. Both therapies were implemented in weekly sessions of 2 h for a month. The results showed how both therapies led to a reduction in caregivers' burden and increased positive emotions.

A few years later, this study was replicated in France following a similar design, this time including a control group [17]. They implemented the same non-pharmacological treatments: music therapy and culinary activities. Both therapies took place in 1-hour sessions twice a week, for a period of one month. As before, their results showed a reduction in caregivers' burden. These results coincided with those found in an experimental study [18] with a control group in Finland, which lasted 9 months. In this case, the researchers compared the results in patients and their caregivers after applying three conditions: regular care, music therapy through the singing of caregivers and music therapy from an electronic device. Their results showed that the singing therapy in which the caregivers had fully participated improved their well-being status.

The results of a more recent experimental study [19] in a residence in the United States reaffirmed these findings. Their designed included an individualized music therapy program for patients that involved caregivers. After the intervention, they observed a decrease in caregivers' burden, thus increasing their quality of life.

### 3.2. Effects of the Influence of Music Therapy on the Symptoms of Alzheimer's Disease

All the articles included in this systematic review addressed the influence of music therapy on the symptoms of Alzheimer's disease, confirming the interest of this topic in the field of research. It proved that there was more information available regarding the given objective.

An experimental study [18] with a control group in patients presenting Alzheimer's in a moderate stage was carried out for 9 months in Finland. Here, besides improvements in the emotional symptomatology and patients' behaviors, music therapy led to benefits in remote memory and orientation in patients assigned to music therapy on electronic devices. In a group subjected to singing by caregivers, an improvement in the patient's short-term memory was also observed, thus confirming improvement in patients' cognitive symptoms [19]. Another experimental study [20] carried out in France with a control group, in patients with Alzheimer's in a mild stage, had one group subjected to singing music therapy, and another group with activities related to drawing for 12 weeks. A decrease in anxiety and pain and an improvement in quality of life were observed in both therapies. Improvement regarding memory episodes resulted only in the group subjected to singing therapy.

The results of a study [21] conducted in Spain coincided with these results on cognitive improvement through music therapy. In this study, patients with moderate Alzheimer's received music therapy for 3 months in 36 sessions. The experimental group received music therapy with familiar songs, while the control group received music therapy with unknown songs. The results of the experimental group were more remarkable and significant than those of the control group. They revealed improvements in prospective memory, patients' awareness and affective state. In this line, we found an experimental study [22] carried out

in Spain, which appeared in the impact journal in SJR. Here, patients with Alzheimer's living in a nursing home received the following for two months all together: music therapy, therapy of reminiscence and reality orientation techniques. The main result confirmed that listening to familiar music made patients connect with events from their past, consequently reducing symptoms of anxiety and depression and increasing their orientation in reality.

In a more recent experimental study [23] with a control group in Taiwan, a group of patients with mild-stage Alzheimer's were subjected to music therapy using headphones for 30 min in the morning and 30 min before bed for 6 months. The control group did not undergo any intervention. The implementation of music therapy resulted in a great difference in improvement in the global cognitive profile of the patients subjected to music therapy with respect to those not subjected. In France, an experimental study [24] with a control group went a step further on the influence of music therapy on the cognitive profile of Alzheimer's patients in mild and moderate stages. They had patients learning texts through music therapy for 6 months. The control group participated in text learning but without relating it to music therapy. The main result was that the sung texts were easier to remember than spoken texts, so music therapy facilitated the learning and retention of texts.

An experimental study [25] with a control group in Japan included music therapy, considering active and passive music therapy. Each type was assigned to groups of patients with severe-stage dementia, while the control group did not undergo music therapy interventions. After implementing the sessions, the following short- and long-term memory results were identified: interactive or active music therapy caused greater benefits in the emotional, cognitive, psychological and behavioral symptoms of the patients. Concurring with these results, an experimental study [26] in Italy without a control group designed active and passive stimulation interventions of music therapy for patients in different stages of Alzheimer's. The study included 10 sessions that took place daily for 5 min. The results revealed a more significant improvement in the levels of participation at the verbal level in the patients subjected to active stimulation of music therapy, with respect to those subjected to passive music therapy. However, it is important to notice that these results were found only in some of the patients and not in the entire sample.

A more recent experimental study [27] without a control group in Switzerland had one group of patients with moderate-stage dementia subjected to sung music therapy sessions for 2 weeks. Another group of moderate-stage patients received 2 weeks of music therapy with interactive movements (interactive music therapy). Nurses trained in this therapy taught both groups. The results showed that interactive music therapy led to a higher degree of improvement in well-being than singing music therapy, and depressive symptoms decreased with both therapies.

Regarding the improvement of language disorders through music therapy, the selected articles revealed some controversy. In an experimental study [28] carried out in Spain with a control group, a group of patients with Alzheimer's in mild-moderate stage underwent treatment with music therapy for 6 weeks. Their results showed cognitive improvement in relation to memory, orientation, depression and anxiety in mild and moderate patients. Only patients with Alzheimer's at a moderate stage presented improvement in delusions, hallucinations, agitation, irritability and language disorders. However, in a more recent experimental study conducted with a control group, in which music therapy was applied to patients with moderate-stage Alzheimer's, there was no significant improvement in language disorders [26].

An important dimension of Alzheimer's symptomatology relates to depression, anxiety, stress, agitation, aggressiveness and behavioral and psychological symptoms. In this line, we found a recent experimental study [29] in Spain without a control group, in which a group of patients with mild Alzheimer's received 60-minute music therapy sessions. The results revealed benefits in their symptomatology, with reduced levels of stress, depression and anxiety. Concurring with this line, we found two studies [23] carried out in Taiwan without a control group and another [30] in Germany with a control group. The respective results of these studies showed improvement regarding psychological and behavioral disorders

(decreasing anxiety, agitation and aggressiveness), therefore leading to an improvement in the general state of well-being of the patients. Furthermore, thanks to music therapy interventions, there was a decrease in the dose of anxiolytic and antipsychotic medication.

## 4. Discussion

The realization of this work has allowed a deepening of the scientific evidence regarding the use of non-pharmacological therapy of Alzheimer's disease and its impact on the health of these patients.

Music therapy, a therapeutic discipline that belongs to non-pharmacological therapies, has gained relevance in recent times in the field of neurogeriatrics [31]. The results offered by the 16 selected articles showed that using music therapy reduced the level of stress and anxiety, improved emotional stabilization and promoted positive feelings among patients and their families. The results confirmed that Alzheimer's patients found texts whose contents were sung easier to learn than spoken texts. Music therapy facilitated the learning and retention of texts in memory [24].

Following this idea, it was possible to find other studies that were not included in this article that supported these results. For example, some considered the positive correlation between the reward circuits and the intensity of a shiver (pilomotor reflex) that was induced through music [32,33]. Simultaneously, the circuit associated to fear (tonsil, left hippocampus and ventromedial prefrontal cortex) was inhibited, proving the relationship between music therapy and emotions. People with Alzheimer's may react to some environmental stimuli as if they were under threat [34]. These interactions with concrete stimuli play an important role in the emergence and maintenance of the psychological and behavioral symptoms of dementia. Considering that music has proven its efficacy in reducing symptoms related to anxiety, depression, nervousness and aggressiveness, the effect of regulating the response to stress may explain many of the findings in our results.

Another relevant finding was the benefit of music therapy not only in the patient, but also in the care unit–patient and caregiver–understanding how the cultural dimension can affect care in Alzheimer's patients. In this case, in Mediterranean cultures, the care of patients at home is perpetuated thanks to the creation of this binomial [35] and is not so oriented to the transfer of the patient to specialized care centers.

The results of our study sought to assess the effectiveness of music therapy by integrating the caregiver into the process. This involved them participating actively, either singing music or providing it. Both options offered benefits in orientation, humor, remote memory and patients' general cognitive state. The singing therapy performed by the caregivers also improved the short-term memory and the state of well-being of both patient and caregiver as a care unit [18,27]. This review allowed us to reflect on Alzheimer's disease beyond the biological aspects, examining how it affects psychosocial and family areas, and how this disease can lead to patients' depersonalization, creating continuous grief in their relatives over the years.

Regarding the limitations of the search process, on the one hand, we will look into intrinsic factors of music therapy, and on the other, we will consider other methodological aspects.

Considering the different effects of music therapy, the existing research was very diverse in what concerns the time needed to achieve benefits or the desired effects obtained using this therapy [16,17,29]. In addition to this, some studies did not specify the type of music or instruments used [18,22,24,25].

In terms of design, not all of the revised articles included a comparison of the effects of music therapy with control groups [15,19,27–29]. Some articles failed to separate the specific effect of music therapy from other non-pharmacological ones in patients with Alzheimer's disease [18].

Concerning the size of the sample, we found that, in general, the reviewed publications presented a small and limited number of participants, which did not allow us to ensure either external validity or population representation.

Finally, it should be noted that there was a tendency in the scientific evidence to evaluate this therapy without a control group. This was a handicap regarding the selection of articles and the possibility of establishing comparisons among them. Notwithstanding this, in our results, 10 of the 16 selected articles stemmed from experimental studies with a control group, which sought to assess the efficacy of music therapy as a non-pharmacological therapy.

## 5. Conclusions

Studies support that music therapy benefits Alzheimer's patients, improving their quality of life in the process of this disease, and positively influencing them at the behavioral, cognitive and social levels. The results showed a decrease in the caregiver's burden and an increase in their well-being with the application of music therapy to Alzheimer's patients. This holds both for cases in which caregivers did not implement it and for those in which they participated in the implementation of the therapy.

Studies showed the benefits of music therapy with respect to the symptoms of Alzheimer's patients. They identified an improvement in emotional, psychological and cognitive symptomatology, thus reducing the symptoms of anxiety, depression, irritability, aggressiveness, agitation and improving or stabilizing orientation, social interaction, remote memory, self-identification, positive feelings and emotions. Music therapy increased the quality of life of Alzheimer's patients by reducing and stabilizing the symptoms of this disease, therefore improving its evolution.

Concerning the results about the benefits of music therapy in relation to caregivers' burden, our results point at the need of developing this line of research further, as less data was found on this topic.

**Author Contributions:** Conceptualization, A.B.-P. and E.B.G.-N.; Formal analysis, A.B.-P., M.C.-R. and E.B.G.-N. Research and analysis, A.B.-P., M.C.-R. and E.B.G.-N. Writing and preparation of the first draft, A.B.-P. and E.B.G.-N. Drafting, A.B.-P., M.C.-R. and E.B.G.-N. All authors have read and agreed to the published version of the manuscript.

**Funding:** This research received no external funding.

**Institutional Review Board Statement:** Not applicable.

**Informed Consent Statement:** Not applicable.

**Data Availability Statement:** Not applicable.

**Conflicts of Interest:** The authors declare no conflict of interest.

## Appendix A

**Table A1.** Analysis and synthesis of the articles.

| Author | Country/Year | Type of Study | Objectives | Sample | Main Findings | Theme | Databases |
|---|---|---|---|---|---|---|---|
| De la Rubia, J.E.; García, M.P.; Cabañés, C.; Cerón, J.J.; Sancho, S. [21] | Spain 2016 | Analytical, quasi-experimental and prospective without control group | To evaluate the effectiveness of the implementation of a short music therapy protocol as a tool to reduce stress and improve the emotional state in patients with Alzheimer's disease. | $n = 25$ Alzheimer's disease patients with mild criteria aged 65 years or older. | The application of music therapy reduced levels of stress, depression and anxiety | B | PUBMED |
| Samson, S.; Clément, S.; Narme, P.; Schiaratura, L.; Ehrlé, N. [9] | France 2015 | Experimental study without control group | To determine the impact of music therapy and other non-pharmacological therapies such as culinary activities on memory management and behavioral disorders in patients with severe criteria Alzheimer's and other dementias living in a residence and the impact on their caregivers. | $n = 14$ Study 1, patients with Alzheimer's disease with moderate-severe criteria and other dementias. $n = 48$ Study 2, with the same type of patients but a better-controlled study. In both the two therapies are applied. | In study 1, the application of music therapy improved the patient's emotional functioning more than culinary activities. In study two, which was better controlled, both therapies showed benefits in behavioral disorders and improvement in emotional stability, in addition to enhancing positive emotions in patients and reducing caregiver burden. It was concluded that both therapies were beneficial. | A,B | PUBMED |
| Gómez, M.; Gómez, J. [20] | Spain 2015 | Experimental study without control group | To determine the improvement of the clinical profile of Alzheimer's patients undergoing music therapy. | $n = 42$ Middle-aged patients with mild or moderate Alzheimer's disease. | A great improvement in orientation, memory, depression and anxiety levels was perceived in patients with Alzheimer's in both stages, in addition to improvement in delusions, hallucinations, agitation, irritability and language disorders in patients with moderate-stage Alzheimer's. | B | PUBMED |

**Table A1.** *Cont.*

| Author | Country/Year | Type of Study | Objectives | Sample | Main Findings | Theme | Databases |
|---|---|---|---|---|---|---|---|
| Narme, P.; Clément, S.; Ehrlé, N.; Schiaratura, L.; Vachez, S.; Courtaigne, B.; Munsch, F. [8] | France 2013 | Experimental study with control group | To verify the efficacy of music therapy and the practice of other pleasurable activities such as cooking with respect to the symptoms of Alzheimer's disease and other dementias and caregiver burden. | $n = 18$ Patients with Alzheimer's disease and other dementias in a moderate or severe stage over 65 years of age undergoing music therapy. $n = 19$ Same typology of patients subjected to culinary activities. | With both therapies, an improvement in behavioral and emotional disorders was observed. However, there was no difference with respect to cognitive disorders. The improvement in agitation was most notable in culinary activities therapy. A decrease in caregiver burden was reflected. | A,B | PUBMED |
| Arroyo, E.M.; Poveda J., Gil, R. [13] | Spain 2013 | Experimental study with control group | Examine the impact of listening to music that is familiar to the patient on their own self-awareness. Using a questionnaire for the patient with Alzheimer's before the intervention and after it. | $n = 20$ Patients with Alzheimer's disease with 3 years of evolution over 65 years of age, experimental group. $n = 20$ Control group, with the same type of patients. | The patients submitted to the intervention of this therapy showed a stabilization or improvement in aspects of self-awareness such as personal identity, prospective memory, affective state, representation of the body and introspection. The control group that underwent music therapy but without songs that were familiar to them showed improvement, but not as significant as the other group in these aspects. | B | PUBMED |
| Rita, A.; Manfredi, V.; Schifano, L.; Paterlini, C.; Parente, A.; Tagliavini, F. [23]. | Italy 2017 | Experimental study with control group | Efficacy of music therapy as a complementary non-pharmacological treatment to pharmacological treatment on communication or language impairment in Alzheimer's disease. | $n = 45$ Patients with Alzheimer's disease over 65 years. | No significant improvement was seen in language disorders, complementing both treatments. However, an improvement was observed in the emotional profile of the patients with the complementation of pharmacological and non-pharmacological treatment (music therapy). | B | PUBMED |

| Author | Country/Year | Type of Study | Objectives | Sample | Main Findings | Theme | Databases |
|---|---|---|---|---|---|---|---|
| Onieva M.D., Hernández L., Parra M.L., González M.T., Fernández E. [14] | Spain 2017 | Experimental study with control group | Identify the effects of music therapy and reminiscence therapy jointly through the application of techniques that guide reality. | $n = 10$ Control group patients with Alzheimer's disease Alzheimer's. $n = 9$ Patients undergoing the intervention. All patients reside in a nursing home. | An improvement in the anxiety symptoms of Alzheimer's disease was seen with the combination of these therapies. Listening to music that was familiar to them made them connect with events from their past. So, depression decreased and reality orientation improved. | B | |
| Palisson, J; Roussel, C; Maillet, D; Belin, C; Ankri, J; Narme, P [16] | United Kingdom 2015 | Experimental study with control group | To observe if the verbal learning of Alzheimer's patients is specific to music therapy or independent of it. | $n = 12$ Patients with a probable diagnosis of Alzheimer's over 65 years of age, experimental group. $n = 15$ Patients diagnosed with Alzheimer's, control group. | It was confirmed that sung texts were easier to remember than spoken texts. The non-association of text learning with music facilitated learning but in the very short-term, that is, it was not retained in memory. Music therapy facilitated learning and the retention of texts in memory. | B | |
| Thomas, K.; Baier, R.; Kosar, C.; Ogarek, J.; Trepman, A.; Mor, V. [11] | USA 2017 | Experimental study without control group | Contrast the results regarding psychological and behavioral symptoms, before and after implementing an individualized music therapy program in patients with dementia residing in residences. | $n = 12,905$ Patients with Alzheimer's disease and other dementias. | It was observed that in all patients subjected to the individualized music therapy program, there was an improvement in psychological and behavioral disorders, thus reducing anxiety, agitation and aggressiveness. In addition, the application of this therapy allowed a reduction in the dose of anxiolytic and antipsychotic medication. Finally, an improvement in the burden of the formal caregiver was observed. | A,B | |

**Table A1.** *Cont.*

| Author | Country/Year | Type of Study | Objectives | Sample | Main Findings | Theme | Databases |
|---|---|---|---|---|---|---|---|
| Sakamoto, M.; Ando, H.; Tsutou, A. [17] | Japan 2013 | Experimental study with control group | To observe the difference regarding the beneficial effects on the symptomatology of applying music therapy and not applying it in patients with mild and moderate dementia. Determine the effectiveness of interactive music differentiating it from passive | $n = 13$ Control group, patients with Alzheimer's. $N = 13$ Group with interactive music therapy $n = 13$ Group with passive music therapy. | In the short term, interactive music therapy caused the greatest benefits on emotional symptomatology in Alzheimer's patients. In the long term, benefits were observed in behavioral and psychological symptoms of Alzheimer's; the best results were from interactive music therapy, after passive music therapy and finally from the control group not subjected to therapy | B | SCOPUS |
| Sarkamo, T.; Tervaniemi, M.; Laitinen, S.; Numminen, A.; Kurki, M.; Johnson, J.K. [10] | Finland 2013 | Experimental study with control group | To determine the efficacy of music therapy in two ways (sung by carers or listening to music). Training caregivers for it, in patients with Alzheimer's and other dementias. | $n = 30$ Control group patients with Alzheimer's disease. Alzheimer's under usual care. $n = 30$ Patients subjected to the songs of their informal caregivers/ Relatives. $n = 29$ Patients subjected to music listened to. All patients are over 65 years old. | It was observed that both therapies, both listening to their caregivers singing and listening to music, provided benefits in orientation, mood and remote memory and general cognitive status. Singing therapy by caregivers further improved caregivers' short-term memory and well-being. Finally, listening to music improved the quality of life and mood of the patients. | A,B | SCOPUS |
| Weise, L.; Jakob, E.; Frithjof, N.; Wilz, G. [22] | Germany 2017 | Experimental study with control group | To identify the feasibility and efficacy of individualized interventions related to music therapy by nursing in dementia patients in five residences. | Control group, patients with dementia with standard care. Intervention group, patients with dementia undergoing music therapy. | An improvement in the quality of life of the group subjected to the interventions was identified compared to the control group. In addition, a decrease in the state of agitation and aggressiveness of the patients, and therefore a general improvement in the state of well-being of the patients was observed. | B | SCOPUS |

**Table A1.** *Cont.*

| Author | Country/Year | Type of Study | Objectives | Sample | Main Findings | Theme | Databases |
|---|---|---|---|---|---|---|---|
| Sakamoto, M.; Ando, H.; Tsutou, A. [17] | Japan 2013 | Experimental study with control group | To observe the differences regarding the beneficial effects on the symptomatology of applying music therapy and not applying it in patients with mild and moderate dementia. Determine the effectiveness of interactive music differentiating it from passive. | $n = 13$ Control group, patients with Alzheimer's. $n = 13$ Group with interactive music therapy. $n = 13$ Group with passive music therapy. | In the short term, interactive music therapy caused the greatest benefits on emotional symptomatology in Alzheimer's patients. In the long term, benefits were observed in behavioral and psychological symptoms of Alzheimer's; the best results were from interactive music therapy, after passive music therapy and finally from the control group not subjected to therapy. | B | SCOPUS |
| Sarkamo, T.; Tervaniemi, M.; Laitinen, S.; Numminen, A.; Kurki, M.; Johnson, J.K. [10] | Finland 2013 | Experimental study with control group | To determine the efficacy of music therapy in two ways (sung by carers or listening to music). Training caregivers for it, in patients with Alzheimer's and other dementias. | $n = 30$ Control group, patients with Alzheimer's disease. Alzheimer's under usual care. $n = 30$ Patients subjected to the songs of their informal caregivers/ relatives. $n = 29$ Patients subjected to music listened to. All patients are over 65 years old. | It was observed that both therapies, both listening to their caregivers singing and listening to music, provided benefits in orientation, mood and remote memory and general cognitive status. Singing therapy by caregivers further improved caregivers' short-term memory and well-being. Finally, listening to music improved the quality of life and mood of the patients. | A,B | SCOPUS |
| Weise, L.; Jakob, E.; Frithjof, N.; Wilz, G. [22] | Germany 2017 | Experimental study with control group | To identify the feasibility and efficacy of individualized interventions related to music therapy by nursing in dementia patients in five residences. | Control group, patients with dementia with standard care. Intervention group, patients with dementia undergoing music therapy. | An improvement in the quality of life of the group subjected to the interventions was identified, compared to the control group. In addition, a decrease in the state of agitation and aggressiveness of the patients, and therefore a general improvement in the state of well-being of the patients was observed. | B | SCOPUS |

**Table A1.** *Cont.*

| Author | Country/Year | Type of Study | Objectives | Sample | Main Findings | Theme | Databases |
|---|---|---|---|---|---|---|---|
| Ray, K.; Gotell, E. [19] | Switzerland 2017 | Experimental study without control group | To determine the effects of the use of music and interactive music therapy with movements, with respect to depressive symptoms and well-being in patients with Alzheimer's and other dementias. | $n = 26$ Patients over 65 years of age with Alzheimer's disease and other moderate-stage dementias living in a nursing home. | An improvement in well-being was observed in patients undergoing interactive music therapy with movement. In patients undergoing music therapy in a sung form, an improvement in well-being was identified, but to a lesser extent. Regarding depressive symptoms, a decrease was identified after music therapy interventions. | B | SCOPUS |
| Chien, L.; Ching, L.; Yuan, Y.; Mei, C.; Chung, C.; Chiou, L. [15] | Taiwan 2015 | Experimental study with control group | To identify whether music therapy positively affects the long-term cognitive profile of patients with Alzheimer's in a mild stage undergoing pharmacological treatment. | $n = 35$ Control group, patients with mild-stage Alzheimer's not undergoing music therapy. $N = 52$ Group subjected to music therapy interventions. All patients over 65 years of age. | A difference was determined in terms of global cognitive improvement in patients with Alzheimer's who underwent music therapy interventions. | B | SCOPUS |
| Lancioni, G.E.; Bosco, A.; Caro, M.F.; Singh, N.N.; Green, V.A.; Ferlisi, G.; Zullo V. [18] | Italy 2013 | Experimental study without control group | To assess the effects of active musical stimulation versus passive musical stimulation, with regard to promoting positive participation in Alzheimer's patients. | $n = 11$ Patients with Alzheimer's in different stages between 65 and 95 years old. | In six of the patients, better levels of positive participation were observed through active music therapy. In the remaining patients, there were no differences between the two conditions. It was recommended to use the active music therapy option. | B | CINAHL |
| Pongan, E.; Tillmann, B.; Trombert, B.; Getenet, J.C.; Auguste, N.N. [12] | France 2017 | Experimental study without control group | To identify effects on chronic pain, anxiety, depression and quality of life of choir singing compared to drawing activities in patients with moderate to severe Alzheimer's disease. | $n = 6$ Control group. $n = 31$ Group of patients undergoing singing therapy. $n = 31$ Group of patients undergoing drawing activities. | In both groups, a decrease in pain and anxiety and an improvement in quality of life were observed. Depressive symptoms were only reduced in the drawing group. Memory episodes were only enhanced in the singing group. | B | LILACS |

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
