# Peer review of "Effect of Music Therapy as a Non-Pharmacological Measure Applied to Alzheimer’s Disease Patients: A Systematic Review"

_nursrep, doi:10.3390/nursrep12040076_

Round 1

Reviewer 1 Report

- We suggest that the first word music  be removed from the first sentence of the abstract (effects of music in music therapy), as it is redundant

- Description of conclusions in abstract could be improved - unclear

- We suggest a review of the punctuation given that throughout the text we find incorrectly placed punctuation marks

Author Response

We would like to thank the reviewers for the constructive reading, and the detailed comments on our manuscript. The recommendations and suggested changes have helped us to thoroughly revise our previous version, significantly improving its quality.  

In our revision we have followed each reviewer separately, considering all their respective suggestions. As required, the changes are marked in red, and in what follows we will answer each reviewer individually.  

 REVIEWER 1

We suggest that the first word music be removed from the first sentence of the summary (effects of music on music therapy), as it is redundant.

We have modified the sentence to avoid this redundant effect (lines 13-14).  

- The description of the conclusions in summary could be improved - unclear

In order to improve the clarity of the abstract, we have rewritten both the section referring our Results and the Conclusions (lines 21-26). 

- We suggest a review of the punctuation given that throughout the text we find misplaced punctuation marks

We have revised and significantly edited the entire text, correcting punctuation and adjusting spaces. Considering this as changes, we have marked the new sections in red as required.  

Reviewer 2 Report

Thank you very much for the opportunity to review your article. This topic is very interesting and relevant. Non-complementary therapies represent an area where more studies and evidence are needed. Congratulations.

I have some suggestions that I think could improve your article.

Abstract:

Lines 21 and 22 – …“positively affecting the evolution of Alzheimer´s disease” consider replacing with "smooth the progression of Alzheimer's disease"

Introduction:

Lines 42 to 44 - You need to clarify an idea. Alzheimer's is a type of dementia, is not a cause of dementia nor is it characterized by dementia. So, you shouldn't say what you say: Alzheimer's disease is considered one of the most common causes of dementia.

and

 Alzheimer's disease is a neuro-43 degenerative disorder characterized clinically by dementia

Line 47 - symptomatology. and  -  remove the period between the two words

Lines 48 and 49 - Despite this definition, it is interesting to understand this disease from a point of view that goes beyond the biological. Consider replacing it with “Despite this definition, it is interesting to understand this disease beyond the biological aspects.”

Lines 61 to 64 - Together with an increased longevity among the population we find an increased prevalence of chronic diseases, including chronic mental disorders such as dementia and one of its subtypes: Alzheimer's disease.

I think it would be better if you carried this period to line 41.

Line 67 - non-pharmacological therapeutic measures. Please consider replacing it with "non-pharmacological therapies".

Materials and Methods

Can you include your research questions?

 Line 142 - If you use "from item 2 to item 18", I think it gets less boring

Line 156 – searchthe - separate the words, please

Line 171 – Remove the space between disease and the comma. Same at line 262, after the word therapy.

Figure 1: Legend – Flowchart7, why 7? PRISMA Flowchart?

Author Response

We would like to thank the reviewers for the constructive reading, and the detailed comments on our manuscript. The recommendations and suggested changes have helped us to thoroughly revise our previous version, significantly improving its quality.  

In our revision we have followed each reviewer separately, considering all their respective suggestions. As required, the changes are marked in red, and in what follows we will answer each reviewer individually.  

REVIEWER 2

Lines 21 and 22 – ..."positively affecting the evolution of Alzheimer's disease" consider replacing with "smoothing the progression of Alzheimer's disease"

We have rearticulated the sentence incorporating the suggested nuance regarding the progression of Alzheimer’s disease (lines 21-23).  

introduction:

lines 42 to 44 - You need to clarify an idea. Alzheimer's is a type of dementia, it is not a cause of dementia nor is it characterized by dementia. Therefore, you should not say what it says: Alzheimer's disease is considered one of the most common causes of dementia. and

 Alzheimer's disease is a neuro-43 degenerative disorder clinically characterized by dementia

Thanks for your feedback regarding this differentiation. In order to gain a clearer definition we have specifically referred to Alzheimer´s disease as a type of dementia (lines 44-46).  

Line 47 - symptomatology. and - remove the period between the two words.

We have edited parts in this paragraph, and while doing so we have corrected the punctuation mark (line 51).

Lines 48 and 49 - Despite this definition, it is interesting to understand this disease from a point of view that goes beyond the biological. Consider replacing it with "Despite this definition, it is interesting to understand this disease beyond the biological aspects."

We have reformulated the sentence as suggested (lines 52-53).

Lines 61 to 64 - Along with greater longevity among the population we find a higher prevalence of chronic diseases, including chronic mental disorders such as dementia and one of its subtypes: Alzheimer's disease.

I think it would be better if you take this point to line 41.

We have modified the text according to this suggestion.

Line 67 - Non-pharmacological therapeutic measures. Consider replacing it with "non-drug therapies."

We have rephrased this sentence, substituting the term in line 69.

Materials and methods

Can you include your research questions?

We have explicitly incorporated our research question when opening the first paragraph of this section (lines 107-108).

Line 142: If you use "from element 2 to element 18", I think it becomes less boring

We eliminated the series of numbers from the previous version (line 145).

Line 156 - searchthe - separate the words, please

Line 156 - searchthe - separate the words, please

Line 156 - searchthe - separate the words, please

We have rewritten this paragraph, correcting the lack of space from previous version.  

Line 171: Remove the space between illness and coma. The same on line 262, after the word therapy.

Both spaces have been eliminated in the new version (lines 176, and the line 262 in the previous version, which has also been rewritten).

Figure 1: Legend – Flowchart 7, why 7? PRISMA flowchart?

Here we have considered this suggestion identifying it as figure 1, and specifying it as PRISMA Flowchart. 

Reviewer 3 Report

It is interesting that reviews are made on different treatments related to people with Alzheimer's disease, which provide evidence and help health professionals to acquire knowledge.

As suggestions following the review of this article:

The title should indicate the type of work presented if it is a systematic review.

In addition, a revision of the citation in the text should be made, for example in line 46.

The English translation should be thoroughly revised.

Introduction:

I suggest changing the paragraph from line 67 to line 82, which makes a description of a paper but it would be more convenient to relate it to the work that has been done.

Method:

I recommend mentioning which descriptive observational study protocol has been followed, such as STROBE, and providing a checklist.

I suggest to analyse the quality of the studies chosen for this review.

Results:

The table with the analysed results should be referred to and clarify how many articles have finally been analysed.

Discussion and conclusions:

Although extensive, the discussion should be better organised by paragraphs with similar subject in order not to be confusing.

Author Response

REVIEWER 3

The title should indicate the type of work submitted if it is a systematic review.

Following this suggestion , the title has been modified (line 3).

In addition, a revision of the citation should be made in the text, for example in line 46.

We have revised and modified the references in the text, including this one.

The English translation should be thoroughly reviewed.

The text has been revised and edited thoroughly, changes are marked in red.

Introduction: I suggest changing the paragraph from line 67 to line 82, which makes a description of a job but it would be more convenient to relate it to the work done.

This paragraph has been revised and rewritten to include results in a more agile format (line 145).

Method: I recommend mentioning which descriptive observational study protocol has been followed, such as STROBE, and providing a checklist.

I suggest analysing the quality of the studies chosen for this review.

These two suggestions have been taken into account and we have modified the initial version in this regard. Now we refer to all the measures taken into account in order to assess the quality of the selected papers explicitly, accounting for all the measures in a paragraph  

(CONSORT statement, from Cobos-Carbó A, Augustovski F. Declaración CONSORT 2010: actualización de la lista de comprobación para informar ensayos clínicos aleatorizados de grupos paralelos. Med Clin (Barc). 2011;137(5):213–5.

Results: Reference should be made to the table with the analyzed results and clarify how many articles have finally been analyzed.

This list is included in the Appendix, and it is referred to in the text. Also, we have specified in the text the exact number of articles that were finally selected (line 163).

Discussions and conclusions: Although extensive, the discussion should be better organized by paragraphs with a similar theme so as not to be confusing.

We have edited and rewritten a significant part of these two sections seeking to gain clarity and concreteness. The new sections are marked in red.

We hope that the present version fulfills the required standards, and that with these changes our manuscript can be published in your journal.

We look forward to hearing from you.

Kind regards,

Round 2

Reviewer 3 Report

Thank you for changing the title and abstract of the paper.
In lines 85 to 90 it concludes that recent studies confirm the benefits of music therapy. This paragraph should be redrafted so as to avoid it being a conclusion.
Line 111 begins to develop a methodology following the CONSORT criteria for the analysis of articles. These criteria are usually used to report any clinical trial, not to review the quality of the review, so the recommendations provided for example by the Cochrane guidelines for systematic reviews should be followed.  Although this CONSORT 2010 guideline list may provide guidance, the aim of the article would be different.
In lines 154 to 161, criteria are developed according to the impact factor of the journals, but it is not clear what this kind of criteria is and what it is expected to obtain. Changes should consequently be made to the flowchart.
The article still needs a major editing of the English drafting. You could have recourse to translators provided by the journal.
The references need to be reviewed as they do not appear to conform to the criteria indicated in the journal.

Author Response

REVIEWER 3

Thank you for changing the title and abstract of the paper.

We would like to thank you and the reviewers for the constructive reading, and the detailed comments on our manuscript. The recommendations and suggested changes have helped us to thoroughly revise our previous version, significantly improving its quality. 

In lines 85 to 90 it concludes that recent studies confirm the benefits of music therapy. This paragraph should be redrafted so as to avoid it being a conclusion.

Indeed, your comment is very accurate, we have modified the paragraph and understanding is much easier for the reader.

Line 111 begins to develop a methodology following the CONSORT criteria for the analysis of articles. These criteria are usually used to report any clinical trial, not to review the quality of the review, so the recommendations provided for example by the Cochrane guidelines for systematic reviews should be followed.  Although this CONSORT 2010 guideline list may provide guidance, the aim of the article would be different.

Thank you very much for your wise comment. We have modified the manuscript and included:

In order to evaluate the methodological quality of the documents found and confirm the selection of suitable works, the Critical Appraisal Skills Programme (CASPe) was used to analyze systematic reviews (Cabello López, 2015) and randomized controlled trials (Cabello, 2005). The Queen's Joanna Briggs Collaboration scale (Joanna Briggs Institute Levels of Evidence and Grades of Recommendation Working Party, 2013) was also used for the analysis of descriptive studies

References

1.Cabello, J. . (2005). Programa de lectura crítica CASPe. Leyendo críticamente la evidencia clínica. 10 preguntas para entender un artículo sobre diagnóstico. Guías CASPe de Lectura Crítica de La Literatura Médica. Alicante, I, 5–8. http://www.redcaspe.org/system/tdf/materiales/plantilla_ensayo_clinico_v1_0.pdf?file=1&type=node&id=158&force=%0Ahttp://www.redcaspe.org/herramientas/instrumentos

  1. Cabello López, J. B. (2015). Lectura crítica de la evidencia clínica. 13–17.

3.Joanna Briggs Institute Levels of Evidence and Grades of Recommendation Working Party. (2013). New JBI Grades of Recommendation. Joanna Briggs Institute, October, 1. http://joannabriggs.org/assets/docs/approach/JBI-grades-of-recommendation_2014.pdf

In lines 154 to 161, criteria are developed according to the impact factor of the journals, but it is not clear what this kind of criteria is and what it is expected to obtain. Changes should consequently be made to the flowchart.

Thank you very much again, we have modified the manuscript with your comment and we have noticed how its quality has improved. Thank you very much again.

The article still needs a major editing of the English drafting. You could have recourse to translators provided by the journal.

The research team has revisited the language issues and we have commissioned the services of a translator and interpreter to improve the edition of English, forgive the inconvenience that our previous edition may have caused you.

The references need to be reviewed as they do not appear to conform to the criteria indicated in the journal.

We have reviewed the references and modified those that did not conform to the regulations of the journal. Thank you very much again.
